# GABA: A Key Player in Drought Stress Resistance in Plants

**DOI:** 10.3390/ijms221810136

**Published:** 2021-09-20

**Authors:** Md. Mahadi Hasan, Nadiyah M. Alabdallah, Basmah M. Alharbi, Muhammad Waseem, Guangqian Yao, Xu-Dong Liu, Hany G. Abd El-Gawad, Ahmed Abou El-Yazied, Mohamed F. M. Ibrahim, Mohammad Shah Jahan, Xiang-Wen Fang

**Affiliations:** 1State Key Laboratory of Grassland Agro-Ecosystems, School of Life Sciences, Lanzhou University, Lanzhou 730000, China; hasanmahadikau@gmail.com (M.M.H.); waseem17@lzu.edu.cn (M.W.); yaogq@lzu.edu.cn (G.Y.); liuxd19@lzu.edu.cn (X.-D.L.); 2Department of Biology, College of Science, Imam Abdulrahman Bin Faisal University, P.O. Box 1982, Dammam 31441, Saudi Arabia; nmalabdallah@iau.edu.sa; 3Biology Department, Faculty of Science, University of Tabuk, Tabuk 71491, Saudi Arabia; b.alharbi@ut.edu.sa; 4Department of Horticulture, Faculty of Agriculture, Ain Shams University, Cairo 11566, Egypt; hany_gamal2005@agr.asu.edu.eg (H.G.A.E.-G.); ahmed_abdelhafez2@agr.asu.edu.eg (A.A.E.-Y.); 5Department of Agricultural Botany, Faculty of Agriculture, Ain Shams University, Cairo 11566, Egypt; ibrahim_mfm@agr.asu.edu.eg; 6Key Laboratory of Southern Vegetable Crop Genetic Improvement in Ministry of Agriculture, College of Horticulture, Nanjing Agricultural University, Nanjing 210095, China; shahjahansau@gmail.com; 7Department of Horticulture, Sher-e-Bangla Agricultural University, Dhaka 1207, Bangladesh

**Keywords:** antioxidant enzymes, γ-aminobutyric acid, proline, stomata, signaling molecule

## Abstract

γ-aminobutyric acid (GABA) is a non-protein amino acid involved in various physiological processes; it aids in the protection of plants against abiotic stresses, such as drought, heavy metals, and salinity. GABA tends to have a protective effect against drought stress in plants by increasing osmolytes and leaf turgor and reducing oxidative damage via antioxidant regulation. Guard cell GABA production is essential, as it may provide the benefits of reducing stomatal opening and transpiration and controlling the release of tonoplast-localized anion transporter, thus resulting in increased water-use efficiency and drought tolerance. We summarized a number of scientific reports on the role and mechanism of GABA-induced drought tolerance in plants. We also discussed existing insights regarding GABA’s metabolic and signaling functions used to increase plant tolerance to drought stress.

## 1. Introduction

GABA was discovered in potato tubers and is found in the majority of prokaryotic and eukaryotic species [1]. It is an important component of primary and secondary metabolite synthesis since it is an integral intermediate in nitrogen metabolism and amino acid biosynthesis [2]. In plant growth and development, it functions as an intrinsic signaling molecule. GABA accumulates quickly in plant tissues in response to a variety of abiotic stresses [3].

Drought is a major global challenge in agriculture because the stress it causes disrupts key physiological processes in plants [4,5]. There is a long history of drought responses in plants, with yield reductions are much greater than 25% [6]. Due to long-term drought, plants also suffer from a decrease in relative water content and various metabolic and photosynthetic abnormalities [7,8,9]. However, GABA can help plants withstand a variety of environmental stresses, including drought, salt, and heavy metals. Exogenous GABA application can increase the activity of antioxidant enzymes and the glyoxalase system and is involved in MG detoxification [10].

Regulation of the stomatal pore aperture is a major factor in determining plant productivity and drought tolerance [11,12,13,14,15]. GABA-induced stomatal control has gained particular attention in recent years. Since GABA acts as a signaling molecule in plants, its role in stomatal movement during drought conditions could be important. To the best of our knowledge, GABA-induced stomatal control has not been extensively reviewed in the literature. Therefore, we intend to address recent advances in GABA-induced stomatal behavior and the underlying mechanisms in plants under drought stress. Additionally, we review the existing literature on the biosynthesis, metabolism, and molecular interactions of GABA in plants in response to drought stress. The aim of this review is to explain the mechanisms involved in GABA-mediated enhancement of plant tolerance in drought-stressed plants through antioxidant activity and synergy with other molecules. We then raise questions to be addressed in future research.

## 2. GABA Biosynthesis and Metabolism in Plants

GABA is produced by a metabolic pathway known as the GABA shunt pathway, which was first observed in plants [1]. GABA shunts are essential for both GABA production and the maintenance of optimal GABA levels. It consists of three major reactions that are catalyzed by the cytosolic enzyme glutamate decarboxylase (GAD, E.C.4.1.1.15), SSADH (succinic semialdehyde dehydrogenase) (E.C.1.2.2.16), and mitochondrial enzymes GABA transaminase (E.C.2.6.1.19).

In a nonreversible reaction, which limits the GABA synthesis rate, glutamic acid decarboxylase (GAD) mainly catalyzes the a-decarboxylation of glutamate to GABA [16]. GABA is transformed to succinic semialdehyde (SSA) by GABA-T, which is then transformed to succinate by SSADH [17], as described in Figure 1.

Pyruvate and glyoxylate are required by GABA-T as amino acceptors and produce alanine and glycine, respectively [19]. In plants, GABA-T prefers 2-oxoglutarate to pyruvate, resulting in glutamate recovery for the GAD response found in plants. Thus, succinate can either enter the TCA cycle as an electron donor in the electron transport chain or be released as succinate from SSADH and enter the TCA cycle [20]. However, SSA can be converted to gamma-hydroxybutyrate (GHB) by GHB dehydrogenase (GHBDH), which also occurs in plants, animals, and *E. coli* [16]. In addition to SSADH, other GABA biosynthesis pathway enzymes in several plant forms have been identified; their numbers vary between different species [21,22]. In plants, it is well-established that GAD contains a calmodulin (CaM)-binding domain, which enables in vitro activity at pH 7.0–7.5 to be activated by the Ca^2+/^CaM complex. At acidic pH, GAD activity is influenced by Ca^2+/^CaM and exhibits a distinct pH optimum at 5.8 [23]. Finally, GABA is synthesized from gamma-amino butyraldehyde, which is produced by the combined activities of 4-aminobutyraldehyde dehydrogenase and diamine oxidase (DAO, E.C. 1.4.3.6) [20].

## 3. GABA-Induced Drought Tolerance in Plants

Drought stress is one of the most serious problems that plants face around the world, negatively affecting their growth and development [24,25,26]. Overproduction of reactive oxygen species (ROS) causes damage to plant cellular components in drought-stressed plants [27,28]. Nevertheless, an increasing amount of evidence shows that stress-induced ROS might have a signaling role [28]. ROS have been found to induce proline synthesis under stress conditions [29,30]. Proline, Ala, Glu, and GABA are all metabolites associated with the GABA shunt that accumulate in plants in response to ROS generation [31,32,33]. However, it was proposed that certain ROS, such as H_2_O_2_, can activate signal transduction pathways in plant cells [34]. The decrease in NADPH in the nucleus by H_2_O_2_ was proposed as a way to safeguard cell processes against H_2_O_2_-mediated toxicity. For example, H_2_O_2_ reductions in wheat plants are a precondition for the regulation of redox reactions, driving the expression of certain genes throughout the growth of seeds [35]. Furthermore, ROS promote mitochondrial glutamate dehydrogenase (GDH) activity, providing Glu as a precursor for ornithine and GABA production [36] (Figure 2).

The protective role of GABA against drought-stress-induced oxidative damage has been demonstrated in a variety of plant species (Figure 2). In *Phaseolus vulgaris* L., for example, exogenous foliar application of GABA increased fresh shoot weight, dry shoot weight, and leaf area. Through osmotic adjustment, GABA improved drought tolerance in *Phaseolus vulgaris* L. [37] (Table 1).

Exogenous application of GABA effectively mitigated drought-induced leaf damage in plants, as demonstrated by a considerably increased relative water content and decreased electrolyte leakage and lipid peroxidation [45]. GABA increased drought resistance in plants, as well as the accumulation of amino acids, organic acids, and other osmotic compounds associated with secondary metabolism [41]. Furthermore, exogenous GABA treatment increased GABA transaminase and alpha ketone glutarate dehydrogenase activity in white clover leaves, but glutamate decarboxylase activity was reduced under control and drought conditions, leading to an increase in endogenous glutamate (Glu) and GABA levels [45]. GABA, which acts as a downstream signaling molecule of stress-related transcription factors, such as WRKY, MYB, and bZIP that confer drought tolerance in plants, was shown to increase Ca^2+^-dependent protein kinase 26 (CDPK26) and mitogen-activated protein kinase 1 (MAPK1) by threefold and fivefold, respectively, during drought stress [40].

## 4. GABA and Antioxidant Systems under Drought Stress

In numerous plant cell organelles, such as plasma membranes, peroxisomes, chloroplasts and mitochondria, reactive oxygen species (ROS) are produced under normal and stressful conditions. The major sources of ROS generation under normal light circumstances are chloroplasts and peroxisomes [46]. ROS overproduction is linked to oxidative damage in plants [47,48,49] and is regulated by genotype, phase of growth, and conditions such as drought [50]. Plants respond to ROS by increasing antioxidant defenses [51,52], which involves several compounds, such as carotenoids, proline, anthocyanins, ascorbate, tocopherols, flavonols and amino acids, and enzymes, such as superoxide dismutase (SOD), catalase (CAT), peroxidase (POX), ascorbate peroxidase (APX), monodehydroascorbate reductase (MDHAR), and dehydroascorbate reductase (DHAR). In general, cell membrane integrity is protected by antioxidant enzymes when the production of ROS in cells increases under stress. Ascorbate is a ubiquitous non-enzymatic antioxidant with significant potential for not only scavenging reactive oxygen species but also for influencing a number of key processes in plants under both stress and non-stress situations [51,52]. The application of exogenous GABA improved the activities of CAT, POD APX, MDHAR and GR enzymes and reduced H_2_O_2_ production, causing greater tolerance to drought in plants [10]. However, GABA application had no significant influence on the activities of superoxide dismutase and catalase in perennial ryegrass grown under well-watered or drought conditions [39]. These data imply that by activating glyoxalase and antioxidant pathways, GABA increases tolerance to oxidative stress caused by abiotic stressors such as drought.

## 5. GABA-Induced Stomatal Regulations under Drought Stress

The regulation of the stomatal aperture is a critical driver of plant productivity and drought resilience and has a significant impact on climate due to its influence on global carbon and water cycling [15,53]. The stomatal pores are marked by a pair of guard cells. Cell volume and pore aperture are regulated by the fine control of ion and water transport across guard cell membranes via transport proteins in response to opening and closing signals, such as light and dark [15,49,54] (Figure 3).

Stomatal guard cells, due to their essential functions in plant cell signaling and their capacity to respond to and incorporate multiple stimuli, have emerged as a prominent model system for elucidating numerous important pathways that regulate plant biotic and abiotic stress tolerance [56,57]. The effects of GABA on stomata have been observed in a variety of plants, including dicot and monocot crops. Recently, GABA and the ALMT1 protein have been discovered to influence plant development through the binding of GABA to the ALTM1 transporter [58,59]. The authors characterized the relationship as typical of ALMT family proteins and speculated that it might alter stomatal movement. ALMT6, ALMT9, and ALMT12 proteins are located in the vacuolar and plasma membranes, and play a role in regulating stomatal movement by enabling malate influx and efflux [60,61]. Meyer et al. (2011) [62] suggested the existence of a cofactor (cytosolic or vacuolar) that binds to and controls the action of the ALMT6 protein since removing Ca^2+^ from the activated protein had no effect on activity. In recent years, Xu et al. 2021 [55] revealed a GABA signaling route in plants, which is described by the simplified models provided in Figure 3. This work suggested that cytosolic GABA signals generated by GAD2 affect stomatal opening, water-use efficiency (WUE), and drought tolerance by negatively controlling the activity of ALMT9. ALMT9 is a key component for the transduction of GABA signals in cells under well-watered and drought conditions (Figure 3). However, the effect of GABA on stomatal closure under drought is still not clear. In addition, preceding research concluded that GABA regulation of stomata might be caused by light, dark, and low signal intermediate concentrations.

## 6. GABA and Polyamines Interrelationships under Drought Stress

Polyamines (PAs) are found in most eukaryotic cells and mainly include Put, Spd, and Spm, which are classified as free, conjugated, or bound in higher plants [63]. Under stressed conditions, other plants display an increase in PA biosynthesis and catabolism [63,64]. PA is widely used in living organisms [65] and is projected to be a signaling molecule that plays pivotal roles throughout physiological processes in regulating developmental stages, such as embryonic differentiation [66], cell death responses [67], fruit ripening [68], and seed germination [69]. PA biosynthesis is caused by ornithine or arginine decarboxylation, which is mediated by the decarboxylase (ORD) of ornithine or the decarboxylase of arginine, respectively (Figure 4).

The importance of GABA in plant tolerance and its relationship with PAs have been thoroughly documented [63]. PA catabolism generates GABA through CuAO or PAO catalysis, which is a direct link between GABA and PAs. Yang et al. (2013) [70] showed that under hypoxic stress, PA degradation provided approximately 30% of the GABA content in fava beans (*Vicia faba*). Three enzymes (ADC, ODC, and SMADC) involved in the production of Pas, and two enzymes (CuAO and PAO) involved in the breakdown of Pas, were activated under drought conditions [45]. The findings showed GABA boosted PA synthesis but, at the same time, inhibited PA catabolism, which increased the overall PA content in GABA-treated plants grown under drought. This result showed that enhanced endogenous GABA acted as negative feedback for the breakdown of PAs, which was in accordance with the findings of Wang et al. (2014) [64].

PA accumulation in various plants was observed under environmental stresses, and the critical roles of PA were validated in plants [13]. Similarly, the accumulation of GABA was reported to reduce the oxidative damage produced by ROS, resulting in increased oxidative stress tolerance [71]. In soybean plants, drought stress led to the accumulation of GABA. This was accomplished by boosting the activity of diamine oxidase (DAO) to increase PA breakdown, which suggests that PA regulates GABA accumulation-driven tolerance responses [72]. In addition, muskmelon (*Cucumis melo*) exposed to salt stress exhibited degradation and endogenous GABA accumulation [73]. Following exogenous GABA treatment, the accumulation of Put, Spm, and Spd was documented by blocking PA degradation (Figure 4). These results indicated that GABA might act as an additional modulator of plant tolerance by modulating the metabolism of PAs. Under conditions of stress, GABA boosts PA levels by blocking its own synthesis. However, the exact mechanism through which this regulatory system is induced in plants remains unknown.

## 7. Conclusions and Future Perspective

In plants, GABA plays a key role in the control of abiotic stress tolerance. GABA use in stressed plants enhances the activities of the GABA shunt; boosts the photosynthetic efficiency of endogenous GABA and antioxidative enzymes; reduces ROS production, MDA, and H_2_O_2_ content; and protects the cell membrane. The majority of studies were conducted under abiotic stress; hence, the significance of the GABA shunt and its metabolites remains unclear, which requires further investigation. GABA plays a major role in the movement of the stomata in plants, especially under drought stress. We think that GABA may have a significant function in the closure and opening of stomata in plants. Substantial work has elucidated the overall function of GABA-mediated stomatal control under drought stress, but more research is needed to clarify the processes behind that control and how it interacts with other elements of stomatal function.

Finally, it is proposed that GABA plays a positive role in physiological regulation when plants are exposed to adverse environmental pressures; thus, manipulating GABA levels could be a viable technique for improving plant stress resistance. This detailed analysis of the synthesis and stress-mitigating role of GABA could be used to increase the amount and quality of agricultural products and contribute to the achievement of sustainable development goals.

## Figures and Tables

**Figure 1 ijms-22-10136-f001:**
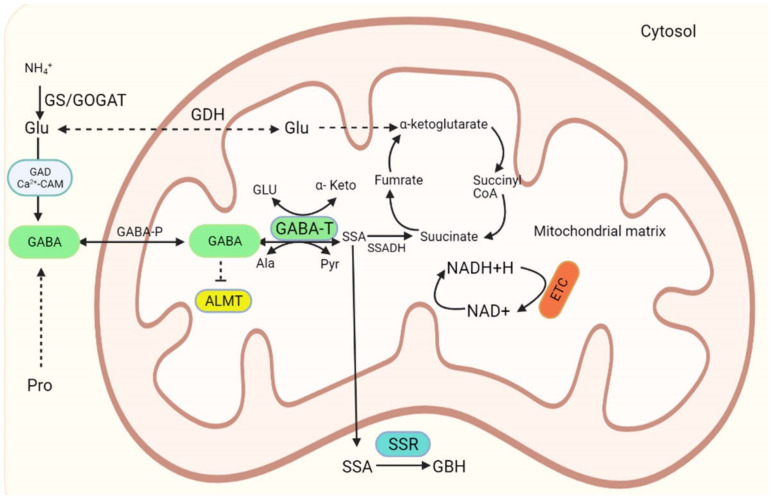
GABA biosynthesis in plants mediated by GABA shunts as adapted from Ramos-Ruiz et al. (2019) [18]. Abbreviations: GAD, glutamate decarboxylase; GABA-P, GABA permease; GABA-T, GABA transaminase; ALMT, aluminum-activated malate transporter; Glu, glutamate; Ala, alanine; Pyr, pyruvate; SSADH, succinic semialdehyde dehydrogenase; ETC, electron transport chain; SSR, succinic semialdehyde reductase; SSA, succinic semialdehyde; GBH, gamma-hydroxybutyric acid.

**Figure 2 ijms-22-10136-f002:**
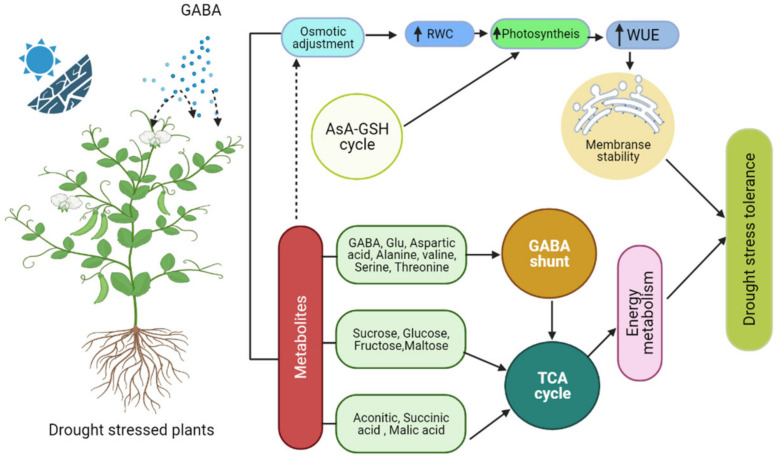
A proposed model demonstrates how GABA protects plants from drought stress.

**Figure 3 ijms-22-10136-f003:**
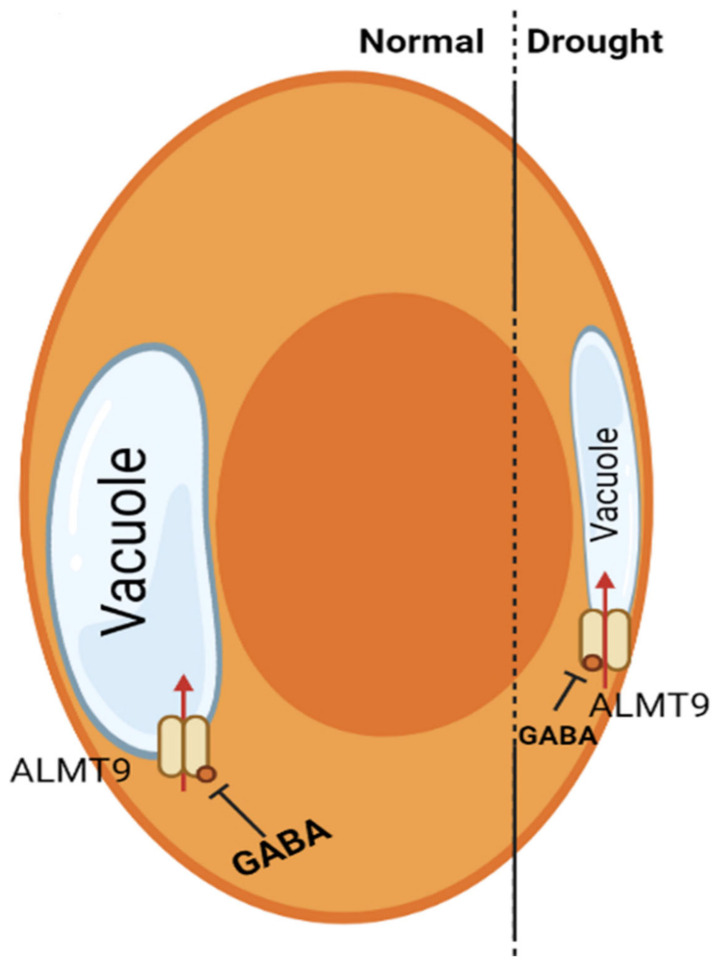
A model for the regulation of water-use efficiency via GABA-mediated signaling adapted from Xu et al. (2021) [55]. During severe drought, production and storage of leaf GABA decreases ALMT9-mediated vacuolar anion absorption into guard cells, which requires the amino acids F243/Y245 (red dot) (right guard cell of pair). Compared to guard cells under normal conditions in the light, this reduces stomatal opening and pore apertures and increases plant water-use efficiency (WUE) during drought stress (left guard cell of pair).

**Figure 4 ijms-22-10136-f004:**
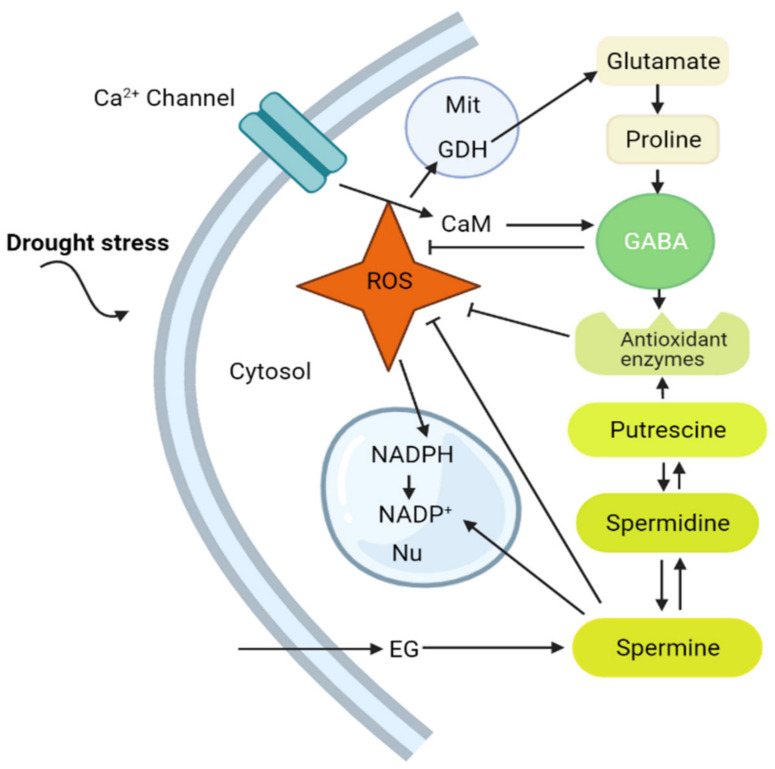
The interrelationship between GABA and PAs mediating ROS in plants. ROS stimulate the activity of GDH in the mitochondria, resulting in the synthesis of Glu, a precursor to GABA. GABA accumulates in response to drought stress via CaM protein-mediated activation of the GAD enzyme and Ca^2+^ signaling. When drought stress occurs, PA biosynthesis increases, which results in GABA biosynthesis. GABA and PAs boost antioxidant enzyme (AE) activity, impairing the synthesis and function of ROS in plant cells. Abbreviations: Ca^2+^, calcium; CaM, calmodulin protein; ROS, reactive oxygen species; GDH, glutamate dehydrogenase; Mit, mitochondria; NADP, nicotinamide adenine dinucleotide phosphate; Nu, nucleus; EG, exogenous GABA.

**Table 1 ijms-22-10136-t001:** GABA-mediated growth, photosynthetic traits, antioxidant defense, and osmoregulation in a wide range of plant species under drought stress.

Species	Stress	GABA Treatment	Effect	Outcome	References
*Phaseolus* *vulgaris* L	Drought (semiarid conditions)	0.5, 1.0, and 2.0 mM (foliar application)	Increased leaf area, fresh and dry shoot weight, and improved osmotic adjustment, membrane permeability, uptake of nutrients, and antioxidant defense	Increased drought tolerance of *Phaseolus* *vulgaris* L.	[37]
*Coix lacryma-jobi* L.	Withholding water	20.0 mM (foliar application)	Preserved the electron transport chain and minimized oxidative damage caused by reactive oxygen species	Mitigated the deleterious effects of drought in *C. lacryma-jobi* plant leaves	[38]
Ryegrass *(Lolium perenne)*	Withholding water	50.0 or 70.0 mM (foliar application)	Reduced lipid peroxidation and electrolyte leakage and improved relative water content (RWC) and antioxidant activity	Alleviated drought stress in ryegrass seedlings	[39]
Creeping bentgrass *(Agrostis stolonifera**)*	Withholding water (soil volumetric water content declined to 7%)	0.5 mM (foliar application)	Increased turf quality, leaf water content, cell membrane permeability, photosynthetic pigments, and expression of CDPK26, MAPK1, ABF3, WRKY75, MYB13, HSP70, MT1, 14-3-3	Significantly improved plant tolerance to drought stress	[40]
Creeping bentgrass *(Agrostis stolonifera)*	Withholding water	0.5 mM (foliar application)	Increased amino acid (GABA, glycine, valine, proline, 5-oxoproline, serine, threonine, aspartic acid, and glutamic acid) and organic acid (malic acid, lactic acid, gluconic acid, malonic acid, and ribonic acid) accumulation	Enhanced drought tolerance	[41]
Sunflower (*Helianthus annuus* L.)	50% field capacity of drought stress	2.0 mg L^−1^ (foliar application)	Increased plant height, fresh and dry weight of shoot and root; improved osmolyte metabolism, expression of genes, and antioxidant enzyme activity	Effectively alleviated drought-induced oxidative stress	[10]
Cumin (*Nigella sativa* L.)	Three irrigation treatments (irrigation after 50, 100, and 150 mm evaporation based on evaporation from class A pan)	0, 0.5, 1.0, and 2.0 mg L^−1^ (foliar application)	Significantly improved chlorophyll content and antioxidant activity	Improved growth and productivity	[42]
*Matricaria recutita* L.	Two levels of 100 (mild stress) and 150 mm (severe stress) evaporation from class A pan	50.0 mM (foliar application)	Positive proline response to severe and mild stress in the presence of GABA	Improved drought tolerance	[43]
Black pepper *(Piper nigrum* L.)	PEG (polyethylene glycol 6000; 10% *w*/*v*)	2.0 mM (GABA-primed black pepper)	Reduced wilting percentage Increased leaf RWC and antioxidant enzyme activity; more rapidly decreased cell osmotic potential; reduced lipid peroxidation rate; significantly decreased inhibition of photosynthetic and mitochondrial activity	Enhanced drought stress tolerance	[44]
White clover (*Trifolium repens*)	15% PEG-induced drought stress	8.0 mM (pretreated plants with GABA in roots)	Increased activities of GABA transaminase and alpha ketone glutarate dehydrogenase; potential GABA-promoted production of polyamines (PAs) and inhibition of their metabolism	Improved drought tolerance of white clover	[45]

## Data Availability

Not applicable.

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
