# Peer review of "GABA: A Key Player in Drought Stress Resistance in Plants"

_ijms, 2021, doi:10.3390/ijms221810136_

Round 1

Reviewer 1 Report

The manuscript deals with a relevant subject to Special Issue "Drought-Stress Induced Physiological and Molecular Changes in Plants". The review presents an important set of physiological and biochemical GABA effects on plants growing under drought stress. On the other hand, the paper is well written and it is supported by a set of very robust references. The over comment is that the manuscript is suitable for publication in International Journal of Molecular Sciences after a minor revision, particularly related with:

  1. Title: “GABA: A key player in drought stress tolerance in plants”. Please, reconsider the word "resistance" instead of "tolerance". The word "resistance" is broader. In all GABA effects reported in this review, in addition to “tolerance”, there are also “avoidance” indicators.
  2. Keywords: The authors should avoid the same words in the title and here. I suggest γ-Aminobutyric acidinstead of GABA in keywords. Keywords should be organized alphabetically.
  3. In this review the authors suggest that “GABA tends to have a protective effect against drought stress in plants by increasing osmolytes and leaf turgor, and reducing oxidative damage via antioxidant regulation.” Next, it is stated that “Guard cell GABA production is absolutely essential, and has the potential benefit of reducing stomatal opening and transpiration”. These two ideas expressed throughout the review seem contradictory. Is it really so?
  4. Table 1: This table is not clear and lacks relevant information. Per example, the description of each item should be more concise and clearer (in some cases the application of GABA is expressed in mg L-1, in others in mM; What do the authors mean by “Irrigation after 150mm evaporation”?). Authors should always indicate how GABA was applied (foliar or other way), as well as the developmental status of the plants during the experiment. As outcome, they should also include a quantitative indicator of growth (fresh and dry mass, total leaf area, etc.).
  5. Page 7, line 192: “… and has a strong impact on the climate because …” - This is unclear. Please, re-phrase it.
  6. Page 10, line 309: “… GABA has an impact on mineral uptake in plants …” – Please, introduce more objectivity into this statement. As it stands, it is too vague.
  7. According to the various studies already carried out, what is the most suitable plant growth phase for the application of GABA? Will a single application of GABA be enough to promote greater resistance to drought?

Author Response

The manuscript deals with a relevant subject to Special Issue "Drought-Stress Induced Physiological and Molecular Changes in Plants". The review presents an important set of physiological and biochemical GABA effects on plants growing under drought stress. On the other hand, the paper is well written and it is supported by a set of very robust references. The over comment is that the manuscript is suitable for publication in International Journal of Molecular Sciences after a minor revision, particularly related with:

Response: Thank you so much for your nice valuable assessments and comments to our manuscript.

Title: “GABA: A key player in drought stress tolerance in plants”. Please, reconsider the word "resistance" instead of "tolerance". The word "resistance" is broader. In all GABA effects reported in this review, in addition to “tolerance”, there are also “avoidance” indicators.

Response: We have revised the title based on your suggestions. We have added “resistance” instead of “tolerance”. Title now reads: “GABA: A key player in drought stress resistance in plants”.

Keywords: The authors should avoid the same words in the title and here. I suggest γ-Aminobutyric acid instead of GABA in keywords. Keywords should be organized alphabetically.

Response: Thank you. γ-Aminobutyric acid has been added in the keywords instead of GABA. In addition, keywords have been arranged alphabetically. Please check the keywords.

In this review the authors suggest that “GABA tends to have a protective effect against drought stress in plants by increasing osmolytes and leaf turgor, and reducing oxidative damage via antioxidant regulation.” Next, it is stated that “Guard cell GABA production is absolutely essential, and has the potential benefit of reducing stomatal opening and transpiration”. These two ideas expressed throughout the review seem contradictory. Is it really so?

Response: Regulation of the stomata is one of the key factors in determining plant productivity and drought tolerance. We have shown that guard cell GABA production is absolutely essential, and has the potential benefit of reducing stomatal opening and transpiration, and is adequate to control the release of tonoplast-localised anion transporter, resulting in higher water use-efficiency and better drought tolerance. Therefore, along with other mechanisms GABA helps plant to improve drought stress tolerance.

Table 1: This table is not clear and lacks relevant information. Per example, the description of each item should be more concise and clearer (in some cases the application of GABA is expressed in mg L-1, in others in mM; What do the authors mean by “Irrigation after 150mm evaporation”?). Authors should always indicate how GABA was applied (foliar or other way), as well as the developmental status of the plants during the experiment. As outcome, they should also include a quantitative indicator of growth (fresh and dry mass, total leaf area, etc.).

Response: Thank you so much for bringing this point. We have revised the table 1 based on your comments and suggestions. The units vary because previous studies reported varied concentrations of GABA (mM, mgL-1) in various crops for application as a treatment in plants.

Water deficit or drought was imposed by the authors using the evaporation pan method, which is employed to hold water during observations (Reference-42) and we have revised the sentence “irrigation after 150mm evaporation”. Please check the table 1 now.

Methods of application (foliar/other way) were added in the table 1. Moreover, developmental status (increase/decrease) and growth parameters (fresh and dry mass, total leaf area, etc.) were also added in the table.

Page 7, line 192: “… and has a strong impact on the climate because …” - This is unclear. Please, re-phrase it.

Response: Thank you. The sentence has been revised and rephrased. Please check line-159-161.

Page 10, line 309: “… GABA has an impact on mineral uptake in plants …” – Please, introduce more objectivity into this statement. As it stands, it is too vague.

Response: As the sentence not clear, we have deleted and replaced with new line. Please check the Line-273-275.

According to the various studies already carried out, what is the most suitable plant growth phase for the application of GABA? Will a single application of GABA be enough to promote greater resistance to drought?

Response: To our knowledge from past literature, the majority of experiments were conducted on seedlings in drought-stressed conditions. GABA was found to be helpful in plants under drought stress after a single treatment.

Reviewer 2 Report

A comprehensive review of the effects of GABA and its role in plant stress response.  In places, a bit disorganised. They end their introduction with the statement “to raise questions that will be addressed in future research” but say very little about future research

Their references to drought tolerance could be better selected. Rather than using very recent references, they should be looking back in the literature for more classical papers and reviews.

They have little on ascorbate and its role in controlling Redox and ROS in plants. is there no literature and is it an area for future research?

Line 39.  There is a long history of drought responses in plants and yield reductions are much greater than 25%

The amino receptors should be included in fig 1

Lines 85 to 89 is unclear

Line 90. Catyaylysed is wrong.

I don’t see how fig 2 demonstrates anything except a pretty picture. Why does the Asc-GSH cycle feed into leaf senescence? It would be better feeding into photosynthesis

Paragraph starting line 133. This repeats what is in table 1 and is not needed. Ditto paragraph starting line 172

Line 134. How does GABA improve drought?

Line n167 you don’t need ‘enzyme’. Where is ascorbate?

Line 240. Gaba binding to the ALTM1 transporter would be better than physical content

Line 260. Sentence unclear

Line 304. What does ‘positive modulation mean’? Increase? I would have thought GABA supplementation would increase GABA by direct action, ie uptake?

Author Response

A comprehensive review of the effects of GABA and its role in plant stress response.  In places, a bit disorganised. They end their introduction with the statement “to raise questions that will be addressed in future research” but say very little about future research

Response: Thank you for bringing this point. In the introduction section, we have described aim of this review including their mechanisms. Regarding future research, we have described in details in the section 7 (Conclusion and Future perspective). Text now reads-57-60, and 271-286

Their references to drought tolerance could be better selected. Rather than using very recent references, they should be looking back in the literature for more classical papers and reviews.

Response: Thank you. We have added some highly cited classical papers and reviews in the introduction section based on your suggestions. Please check the reference sections-Ref.no-4-5.

They have little on ascorbate and its role in controlling Redox and ROS in plants. is there no literature and is it an area for future research?

Response: Thank you. We have added new lines regarding ascorbate. Please check Line-149-152.

Line 39.  There is a long history of drought responses in plants and yield reductions are much greater than 25%

Response: Thanks. We have added this suggested line. Please check Line-42-43.

The amino receptors should be included in fig 1

Response: Thank you. Fig. 1 has been revised based on your suggestions.

Lines 85 to 89 is unclear

Response: Thanks. We have revised the lines 85 to 89 and deleted unnecessary lines.

Line 90. Catyaylysed is wrong.

Response: We have revised Line 90 and deleted Catyaylysed. Please check Line-93-95.

I don’t see how fig 2 demonstrates anything except a pretty picture. Why does the Asc-GSH cycle feed into leaf senescence? It would be better feeding into photosynthesis

Response: Thank you. We have revised the figure 2 based on your suggestions. We have showed that Asc-GSH cycle feed into photosynthesis instead of leaf senescence according to your suggestions.

Paragraph starting line 133. This repeats what is in table 1 and is not needed. Ditto paragraph starting line 172

Response: Thanks for your suggestions. We have deleted unnecessary lines those descriptions are already in Table 1.

Line 134. How does GABA improve drought?

Response: We have rephrased the sentence and added new Line “Exogenous application of GABA effectively mitigated drought-induced leaf damage in plants, as demonstrated by considerably increased relative water content and decreased electrolyte leakage and lipid peroxidation”. Please check Line-124-126

 Line n167 you don’t need ‘enzyme’. Where is ascorbate?

Response: Ok. Thank you. The word “enzyme” has been deleted and ascorbate has been added in the line. Please check Line-142-144

Line 240. Gaba binding to the ALTM1 transporter would be better than physical content

Response: Thank you. Physical contact has been replaced by GABA binding to the ALTM1 transporter. Please check Line-208-209.

Line 260. Sentence unclear

Response: Thank you. We have deleted this unnecessary sentence.

Line 304. What does ‘positive modulation mean’? Increase? I would have thought GABA supplementation would increase GABA by direct action, ie uptake?

Response: We have revised and deleted line 304 and added new line. Please check Line-207, Line 273-275.